# Determination of Heavy Metal Ions in Infant Milk Powder Using a Nanoporous Carbon Modified Disposable Sensor

**DOI:** 10.3390/foods12040730

**Published:** 2023-02-07

**Authors:** Han Chen, Yao Yao, Chao Zhang, Jianfeng Ping

**Affiliations:** Laboratory of Agricultural Information Intelligent Sensing, School of Biosystems Engineering and Food Science, Zhejiang University, Hangzhou 310058, China

**Keywords:** nanoporous carbon, heavy metal ions, electrochemical detection, milk powder, stripping voltammetry

## Abstract

Due to the risk of heavy metal pollution in infant milk powder, it is significant to establish effective detection methods. Here, a screen-printed electrode (SPE) was modified with nanoporous carbon (NPC) to detect Pb(II) and Cd(II) in infant milk powder using an electrochemical method. Using NPC as a functional nanolayer facilitated the electrochemical detection of Pb(II) and Cd(II) due to its efficient mass transport and large adsorption capacity. Linear responses were obtained for Pb (II) and Cd(II) in the range from 1 to 60 µg L^−1^ and 5 to 70 µg L^−1^, respectively. The limit of detection was 0.1 µg L^−1^ for Pb(II) and 1.67 µg L^−1^ for Cd(II). The reproducibility, stability, and anti-interference performance of the prepared sensor were tested as well. The heavy metal ion detection performance in the extracted infant milk powder shows that the developed SPE/NPC possesses the ability to detect Pb(II) and Cd(II) in milk powder.

## 1. Introduction

Nowadays, man-made and natural disasters, including the universal application of pesticides and fertilizers, mining, smelting, fossil fuel burning, forest fires, and volcanic eruptions [1], have led to serious heavy metal contamination with the development of industry and agriculture. Heavy metal ions (HMIs) have been exposed to food, plant, soil, natural water, and many other sources. With the transmission of the food chain, heavy metal pollution eventually enters the human body, which is harmful to body health. Some studies have shown that the toxicity of HMIs can lead to serious diseases, such as encephalopathy, cardiovascular disease, organ failure, neurogenic disease, and cancer [2]. Recently, research has shown that the exposure of cows to the contaminated environment around steel processing units or thermal power areas can increase the concentration of HMIs in milk [3]. Milk, an important food of animal origin, has most of the nutrients necessary for a healthy diet, especially for children [4]. Rich nutrients in milk powder, such as vitamins and mineral substances, are essential for a baby’s development [5]. Except for the necessary mineral elements, ions in milk products, such as Pb(II), Cd(II), and other HMIs, can lead to toxic effects on babies even at low concentrations. Moreover, several reports have demonstrated the intake of HMIs by infants through milk powder [6,7], which may bring serious consequences to infant health. Thus, it is necessary to develop methods to quantify HMIs in infant milk powders.

Compared with conventional methods for the detection of HMIs, electrochemical techniques that take advantage of the portable, rapid, low-priced, and inherent electrochemical activity of HMIs [2] provide a better alternative for their detection. The comparison of HMI detection with electrochemical methods and other widely used techniques is shown in Appendix A [8,9,10,11,12,13,14,15,16,17,18,19,20,21,22]. The screen-printed electrode (SPE) has received wide attention for its intrinsic properties, such as small size, light weight, ease of mass production, and low cost [23], which makes it suitable for on-site rapid detection compared with the traditional glassy carbon electrode. However, the non-conductive substances in the carbon inks could lead to poor electrochemical performance [2]. Thus, various nanomaterials were used on SPEs to improve the electrochemical properties.

The application of nanomaterials in electrochemical sensing has received extensive concern due to their specific chemical, physical, and electronic properties, including the large specific surface area, quick mass-transport rate, and ability to speed up electron transfer [24]. Many studies have shown that electrodes modified with nanomaterials, such as metallic nanomaterials [25], carbon nanomaterials [26], silicon nanomaterials [27], and biological nanomaterials [28], can effectively enhance the electrochemical detection performance towards HMIs. Among these nanomaterials, carbon nanomaterials have attracted great interest in HMI detection due to the excellent properties of low-cost and large specific surface areas. Nanoporous carbon (NPC), as a new type of carbon nanomaterial, has been applied widely in the field of electrochemical sensing. Notable merits of low density, excellent thermal and chemical stability, hierarchical porosity, efficient mass transport, and large adsorption capacity [29,30] make NPC exhibit satisfactory properties in adsorption, catalyst, and energy-storage applications [29,31,32]. Most importantly, few reports have explored the performance of NPC in the field of electrochemical detection of HMIs.

In this study, the electrochemical performance of NPC in the detection of HMIs (Pb(II) and Cd(II)) was investigated. A screen-printed electrode (SPE) was applied as the sensing substrate, and NPC was used as the modification layer and applied to facilitate the electrochemical detection of the HMIs by promoting the concentration process. Metal ions, such as bismuth, platinum, gold, and mercury, are preferred for the detection of HMIs because these ions have the ability to form an alloy with HMIs for a higher detection sensitivity [33]. Bismuth is widely used due its merits of being environmental-friendly and low-cost [34]; thus, Bi(III) was applied in this work to provide a simple way to enhance the reduction of HMIs by forming a Bi(III)-based alloy film. The material characterization of the NPC and electrochemical performance of the SPE/NPC were investigated. The relevant HMI detection parameters were optimized, and the critical properties for real applications, including the reproducibility, stability, and anti-interference performance, were also evaluated. The results demonstrated that the developed SPE/NPC possesses satisfactory abilities for the detection of HMIs. To further confirm this point, the prepared SPE/NPC was applied in the monitoring of HMI contamination in infant milk powder.

## 2. Material and Methods

### 2.1. Apparatus

The electrochemical experiments were realized on a PalmSens3 Electrochemical Interface (PalmSens BV, Houten, The Netherlands). The commercially fabricated SPE integrated a three-electrode system, including two carbon electrodes and an Ag/AgCl electrode. The concentration of HMIs was measured with inductively coupled plasma mass spectrometry (ICP-MS) (ELAN DRC-e, PekinElmer, Waltham, MA, USA) as the gold standard. The morphology of the NPC was analyzed with a scanning electron microscope (SEM) (HITACHI-SU8010, Tokyo, Japan). X-ray diffraction (XRD) characterization was performed on a Bruker D8 Advance (Bruker AXS, Saarbrucken, Germany) to clarify the internal structure of the NPC. The Raman spectrum was realized to characterize the structure of the NPC through a LabRAM HR Evolution Raman microscope system (Horiba Jobin Yvon, Kyoto, Japan).

### 2.2. Chemicals and Reagents

SPEs (No. TE100-RI) were ordered from Chanpu Technology Co., Ltd. (Taiwan, China). NPC bulk powder was procured from Hangzhou Heshi New Material Technology Co., Ltd. (Hangzhou, China). Sodium acetate (≥99.0%) was obtained from Acros Organics. Acetic acid (≥99.7%), bismuth nitrate pentahydrate (≥99.999%), lead nitrate (≥99.999%), cadmium nitrate tetrahydrate (≥99.997%), iron nitrate (≥98%), zinc nitrate (≥99%), D-(+)-Glucose (≥99.5%), D-(-)-fructose, and Sucrose (≥99.0%) were procured from Sigma Aldrich. *N*,*N*-Dimethylformamide (≥99.7%), glutamate (≥99.0%), and magnesium nitrate were procured from Alfa Aesar. Hydrogen peroxide, hydrochloric acid, sodium hydroxide, absolute ethanol, and potassium chloride were produced from the National Pharmaceutical Group Chemical Reagent Co., Ltd. (Shenzhen, China). Potassium hexacyanoferrate and potassium ferrocyanide were procured from the Wenzhou chemical material factory (Wenzhou, China).

### 2.3. Preparation of the SPE/NPC

In this study, the electrode preparation process referred to the reported method [35]. To obtain a homogeneous solution of NPC, 0.02 g of NPC bulk powder was dissolved in 10 mL of ethanol and 10 mL of water with sonication for 30 min. Then, the solution was centrifuged at 3000 rpm for 4 min to remove the precipitate, and the supernatant was then centrifuged at 6000 rpm for 15 min; the precipitate was gathered and dried into powder. Then, the powder was dissolved into *N*,*N*-Dimethylformamide to prepare the NPC suspension (1.0 mg mL^−1^). After the preparation of the NPC suspension, the SPE/NPC was constructed. The SPE was ultrasonically cleaned with DI water; then, the residual water was removed with nitrogen gas. After that, 6 µL of the preparative NPC suspension was dropped onto the working electrode, and the solvent was volatilized in a vacuum-drying oven (STIK, Shanghai, China).

### 2.4. Real Sample Preparations

Five different brands of infant milk powder were bought from a supermarket (Hangzhou, China).

The main ingredients and nutrients of the milk powder from the five different brands are listed in Appendix A. The pretreatment method of the milk powder samples was processed according to the reported steps [36]. Specifically, to ensure the complete dissolution of the milk powder, 2.87 g milk powder was added to 20 mL DI water. Then, 50 µL of H_2_O_2_ (30 wt%) was added into 20 mL of milk to remove the reductive substances including vitamins. After a 15 min sonication, 5 mL of hydrochloric acid (36.5 wt%) and 5 mL of acetic acid (50 wt%) were added into the mixture and treated with ultrasound for 8 min to inactivate the biologically active substances including proteins. To further filter out macromolecular substances, the obtained solution was centrifuged at a speed of 10,000× *g* for 15 min, and the interferents in the liquid supernatant were removed with suction filtration using a cellulose filtering membrane (pore size: 0.22 µm). Finally, a solution of 0.1 M NaOH was used to calibrate the pH value of the liquid to 4.5. The prepared solution was stored at 4 °C before testing.

### 2.5. Electrochemical Detection of Pb(II) and Cd(II)

In this work, the Adsorptive Stripping Square Wave Voltammetry (ASSWV) method was applied to detect the concentration of the HMIs. The testing well was filled with acetic acid buffer first. Bi(NO_3_)_3_ was then added to form an alloy with the HMIs and to facilitate the deposition process. After adding Pb(II) and Cd(II) into the mixing solution, the electrochemical program of the CHI work station was operated. Three steps were involved in the electrochemical detection process of the HMIs. First, the preconditioning process was operated at +0.2 V for 100 s to clean the electrode surface. Then, the electrochemical deposition of the HMIs was performed at −1.0 V for 160 s; during the first 150 s, the process was operated with stirring and then followed with an equilibration period of 10 s. Finally, the Square Wave Voltammetry (SWV) method was used for the stripping determination of the HMIs.

## 3. Results and Discussions

### 3.1. Characterization of NPC

As the SEM images show in Figure 1A,B, the surface morphology of the NPC exhibits a porous structure, which provides abundant active sites for the “in situ” electro-deposition of Bi(III) and the following reduction in HMIs. The Raman spectra was applied to confirm the graphite structure of the NPC (Figure 1C). Two peaks at 1320 cm^−1^ and 1595 cm^−1^, corresponding to the D and G band, can be observed, which are attributed to the defects and sp^2^ graphitic structure of the NPC samples, respectively [37].To further explore the structure and composition of the NPC, the XRD pattern was investigated and is shown in Figure 1D. Obviously, there are two peaks in the testing range, including the one at approximately 24 degrees formed by the reflection of (002) the crystal plane in the graphite sheets and the other one at 43 degrees representing the (100) crystal plane of the sp^2^ carbon. The results indicate the graphite carbon structure in the obtained NPC material [37]. The N_2_ adsorption–desorption isotherms were measured to investigate the pore structure of the NPC material (Appendix A). The Brunauer–Emmett–Teller (BET) surface area of the NPC was 217 m^2^ g^−1^. As observed in the pore-size distribution, the main pore size of the NPC was 26.42 nm.

### 3.2. Electrochemical Characterizations of the SPE/NPC

Electrochemical impedance spectroscopy (EIS) was used to characterize the nanomaterials’ modification of the sensor (Appendix A). The diameter of the high-frequency semicircle in the Nyquist spectra represents the electron transfer resistance (RET) at the electrode surface. When the surface of the SPE was coated with NPC, the RET value decreased from 2106 Ω to 1806 Ω, indicating that the presence of NPC could promote the electron transfer at the electrode interface. Before investigating the stripping performance of the SPE/NPC in the detection of HMIs, the hydrogen evolution process at the electrode surface should be considered due to its deep impact on the stripping results. Therefore, the electrochemical window of the SPE/NPC was measured in acetic acid buffer with the cyclic voltammetry (CV) method. As illustrated in Appendix A, the SPE/NPC exhibits a wide electrochemical window ranging from −1.0 V to 1.5 V, during which the background current is higher than the bare SPE, indicating the conductivity of the SPE increases after modification. The above results demonstrate that the SPE/NPC can measure Pb(II) and Cd(II) with improved conductivity without autointerference peaks within the required voltage range.

The electron transfer processes of the bare SPE and SPE/NPC were also compared by their CV curves in the K_3_[Fe(CN)_6_]/K_4_[Fe(CN)_6_] solution. As illustrated in Figure 2A, a pair of reversible peaks can be found in the curves of the bare SPE and SPE/NPC. Notably, the peak potential separation of the bare SPE is 0.456 V, while the one after modifying with NPC reduces it to 0.152 V, and the peak current value also increases from 8.423 µA to 12.320 µA. It can be concluded that the NPC can effectively promote the process of electron transport on the surface of the SPE. To further confirm this conclusion, we calculated the electro-active surface area of the electrode based on the CV curves. According to the Randle–Sevcik equation [38], the effective electro-active area of the electrode can be calculated as follows: *I*_P_ = 2.69 × 10^5^*AD*^1/2^*n*^3/2^*γ*^1/2^*C*, where *I*_P_ is the anodic peak current (µA), *A* is the electro-active surface area (cm^2^), *D* is the diffusion coefficient (cm^2^ s^−1^), *n* is the number of transferred electrons, *γ* is the potential scanning rate (V s^−1^), and *C* is the concentration (mM) of K_3_[Fe(CN)_6_]/K_4_[Fe(CN)_6_]. Based on this, the electro-active surface area of the bare SPE is calculated as 0.038 cm^2^, while that of the SPE/NPC is calculated as 0.056 cm^2^. The surface area of the modified electrode is approximately 1.5-times higher than the bare electrode. Then, the HMI detection performance of the SPE and SPE/NPC was compared with the ASSWV method, and the corresponding stripping curves are shown in Figure 2B. For the bare SPE, a weak stripping response is observed; on the contrary, the SPE/NPC exhibits enhanced current signals for Pb(II) and Cd(II). Consequently, the HMI detection performance can be significantly improved by introducing the SPE/NPC design with enhanced electrochemical characteristics.

### 3.3. Parameter Optimization

To further improve the detection performance, critical experimental parameters were optimized, including the pH value of the acetic acid buffer, deposition volume of the NPC suspension, concentrations of Bi(III), and deposition parameters such as potential and time; the results are shown in Figure 3 and Appendix A. Since it was widely reported that the acetate buffer solution is the best electrolyte for the detection of HMIs with the ASSWV method [34,36], in this work, we directly optimized the pH value of the acetate buffer for the SPE/NPC. The stripping analysis of the HMIs based on the SPE/NPC was performed in acetate buffer with a pH varying from 4.0 to 6.0. As illustrated in Figure 3A, the current intensity increases with the pH value from 4.0 to 4.5. After that, it presents a decreasing trend. The change in current response with pH value is consistent with Xuan et al. [39]. The stripping current responses of Pb(II) and Cd(II) are closely related to the pH value of the acetic acid buffer; the best stripping performance can be obtained when the pH reaches 4.5. The decreased detection performance at a high pH can be attributed to the hydrolysis of Bi(III), which weakens the following deposition process of the HMIs.

The deposition volume of the NPC suspension on the SPE was also optimized. The stripping analysis of the HMIs was conducted on the SPE/NPCs deposited with different volumes (3, 4, 5, 6, 7, and 8 µL) of the NPC suspension (1.0 mg L^−1^). As the results illustrate in Appendix A, the peak current increases with the deposited volume of the NPC suspension; a maximum peak current is achieved in the electrode deposited with 6 µL of the NPC suspension. Subsequently, the detection response of the SPE/NPC exhibits a decreasing trend as the deposition volume is further increased; this can be explained by the decreased conductivity and stability of the NPC layer on the SPE surface with an excessive thickness. Therefore, 6 µL of the NPC suspension is selected as the optimized deposition volume.

The thickness of the Bi-based alloy film is controlled by the concentration of the added Bi(III) solution; Bi(III) solutions with various concentrations that varied from 0.65 to 0.98 mg L^−1^ were evaluated for the detection of HMIs (Figure 3B). As the concentration of Bi(III) increases, the overall trend of the stripping current first increases and then decreases; the peak current appears at a Bi(III) concentration of 0.81 mg L^−1^. The electrochemical deposition process of HMIs can be promoted effectively through the formation of a Bi-based alloy; however, the mass transfer of the metallic deposition process might be hindered by a thick Bi film formed under a high concentration of Bi(III) [40]. Thus, a Bi(III) solution with a concentration of 0.81 mg L^−1^ is adopted for the stripping analysis of the HMIs.

The deposition parameters, including the deposition potential and deposition time, are also critical for the detection of HMIs. Figure 3C,D show the influences of the deposition parameters. As for the optimization of the deposition potential from −0.8 V to −1.0 V (Figure 3C), Pb(II) and Cd(II) were barely detected with the potential of −0.8 V; the signal begins to strengthen with an increase of deposition potential. While the deposition potential exceeds −1.0 V, the current responses begin to decrease due to the reduction in interfering substances in the solution under high potential. From the results exhibited in Figure 3D, the current response has an obvious increasing trend as the deposition time prolongs, and then, it gradually approaches a flat after 150 s of deposition, which can be explained as the HMIs deposited on the electrode have reached saturation. Accordingly, a depositing time of 150 s at a deposition voltage of −1.0 V was applied in the end.

### 3.4. Electrochemical Determination of Pb(II) and Cd(II)

The determination performance of the developed SPE/NPC for HMIs was studied with the ASSWV method, and the experiments were operated under the optimized parameters discussed above. The stripping response diagrams of Pb(II) from 1 to 60 µg L^−1^ and Cd(II) from 5 to 70 µg L^−1^ are illustrated in Figure 4A, and their corresponding calibration curves are shown in Figure 4B and C, respectively. As expected, the peak current of the HMIs shows a linear correlation with the concentration of the detected HMIs. The corresponding linear regression equations are calibrated as follow: *I*_Pb_ = 0.0177*C*_Pb_ + 0.2496, *I*_Cd_ = 0.0396*C*_Cd_ − 0.1259, and the correlation coefficients are 0.9884 and 0.9958, respectively. Moreover, the detection limits (S/N = 3) of 0.10 μg L^−1^ for Pb(II) and 1.67 μg L^−1^ for Cd(II) can be acquired, respectively. The stripping response diagrams of single variable ions are shown in Appendix A, and the relevant standard curves are attached as well. The corresponding linear regression equations are calibrated as follow: *I*_Pb_ = 0.0376*C*_Pb_ + 0.3293 and *I*_Cd_ = 0.0637*C*_Cd_ − 0.8962. Compared with simultaneous detection of two ions, the response of a single ion is more sensitive for Pb(II) and Cd(II). The possible reason is the competition between the two ions during the process of concentrating on the electrode surface [41]. Furthermore, the detection performance of our method is compared with the reported results and shown in Appendix A [42,43,44,45,46,47]. Notably, the SPE/NPC of our work shows comparable analytical performance in the detection of Pb(II) and Cd(II).

### 3.5. Reproducibility, Stability, and Anti-Interference Performance of the SPE/NPC

To evaluate the reproducibility of the developed SPE/NPC, five electrodes were selected randomly to detect Pb(II) and Cd(II) with a concentration of 30 µg L^−1^. As revealed in Figure 5A, the relative standard deviations (RSD) of the peak current are 6.6% and 10.1% for Pb(II) and Cd(II) (*n* = 5), respectively. The stability of the SPE/NPC was also evaluated with a long-term operation, which means the same electrode was used eight times to measure Pb(II) and Cd(II). The peak currents of the SPE/NPC are recorded in eight consecutive detection times in Figure 5B. There is no obvious change in the stripping responses of the HMIs; the RSD is 4.8% for Pb(II) and 6.9% for Cd(II), respectively, indicating good stability of the prepared electrode under the test environment.

The anti-interference ability is a critical property for sensors in practical use, especially for food safety monitoring since the composition of food is complex and the active components will interfere with the testing process. Interfering substances, including Ca(II), Zn(II), Mg(II), Fe(III), fructose, glucose, sucrose, and glutamate, may exist in infant milk powder. Based on this, the interferences mentioned above were selected for the selectivity evaluation of the SPE/NPC. Interfering substances, including Ca(II), Zn(II), Mg(II), Fe(III), fructose, glucose, sucrose, and glutamate, were added into the solution containing the HMIs. The concentration of Pb(II) and Cd(II) was 30 µg L^−1^, while that of the interfering substances was controlled at 1000 µg L^−1^. After that, the stripping test was performed (Appendix A). After the addition of the interferences, acceptable changes in the peak current compared with the responses of Pb(II) and Cd(II) were observed.

### 3.6. Real Samples Application

The SPE/NPC was applied for the detection of HMIs in infant milk powder to verify the feasibility for practical application. A recovery test was used to assess the performance of our method for real sample analysis. Briefly, 30 µg L^−1^ of Pb(II) and Cd(II) were added into the pre-treated solution of infant milk powder samples; then, the SPE/NPC was applied to detect the HMIs in the testing solution. The stripping curves before and after the addition of the HMIs in the five samples are shown in Appendix A. To evaluate the accuracy of the data obtained from our sensor, a standard method of ICP-MS was used for measuring the ion concentration simultaneously. The recovery result is calculated as follows: *R*_re_ = *C*/*C*_0_, where *C* and *C*_0_ are the concentration of the HMIs tested by our method and the added concentration, respectively. The average recoveries for Pb(II) and Cd(II) were 115.32% and 86.78%, respectively (Table 1), and the testing results of Pb(II) and Cd(II) in infant milk powder acquired with the SPE/NPC was close to the results obtained with spectroscopic techniques. The behavior of single-variable heavy metal ions in real samples are concluded in Appendix A. The average recoveries for Pb(II) and Cd(II) were 83.08% and 102.76%, respectively. Most importantly, the limitations of the standard methods, including expensive costs and professional operators, are avoided in our method. The satisfactory results demonstrate the feasibility of our method for the detection of Pb(II) and Cd(II) in infant milk powder, and it takes advantage of the effective, low-cost, and easy operation in the real sample application.

## 4. Conclusions

In this study, a SPE/NPC was developed as a transducer for the electrochemical detection of HMIs (Pb(II) and Cd(II)). The use of a SPE provides a portable, disposable, and efficient platform for the electrochemical determination. Notably, the unique merits of the graphite carbon phase and porous structures of NPC effectively promote the mass transfer on the sensing surface. Its surface morphology exhibits a porous structure, which provides abundant active sites for the “in situ” electro-deposition of Bi(III) and the following reduction in HMIs. Under optimal conditions, the SPE/NPC sensor offered a linear response ranging from 1 to 60 µg L^−1^ and 5 to 70 µg L^−1^, and satisfactory LOD of 0.1 µg L^−1^ and 1.67 µg L^−1^ for Pb(II) and Cd(II), respectively. After a long-term operation, no obvious change in the stripping responses of the HMIs was observed. For the selectivity evaluation, acceptable changes in the peak current compared with the responses of Pb(II) and Cd(II) were observed after the addition of interferences. Though excellent stability and anti-interference performance were achieved, the reproducibility still needs to be further improved. It is worth noting that there is a certain difference in the standard curves when the two ions are detected simultaneously and separately. The higher sensitivity of single heavy metal ions may imply the competition relationship between the two different ions. The interaction mechanism between the two different ions in this process needs to be further investigated. The average recovery ratio of 115.32% for Pb(II) and 86.78% for Cd(II) was realized in the real application in infant milk powder. Given the excellent electrochemical performance and simple fabrication process, the SPE/NPC developed in our work provides a promising method for the detection of HMIs in milk powder samples.

## Figures and Tables

**Figure 1 foods-12-00730-f001:**
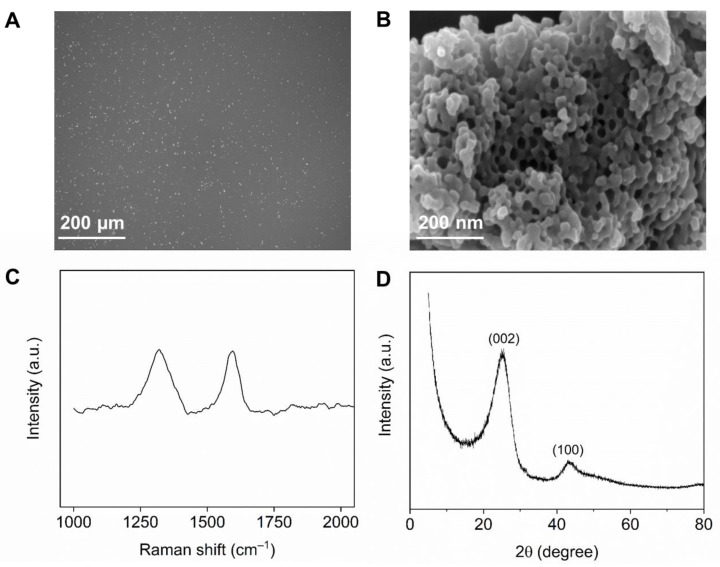
(**A**,**B**) Scanning electron microscope images, (**C**) Raman spectra of the NPC, and (**D**) X-ray diffraction pattern.

**Figure 2 foods-12-00730-f002:**
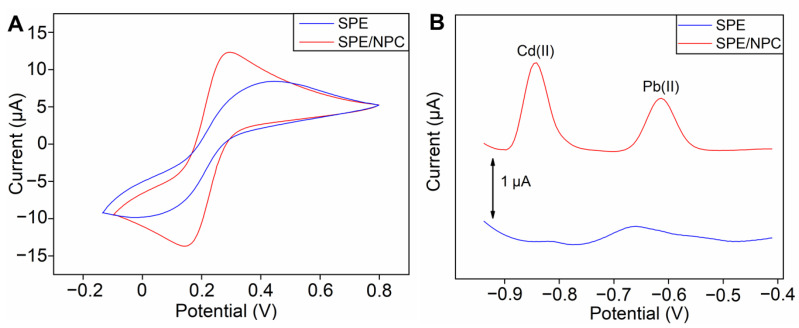
(**A**) Cyclic voltammogram curves of the bare SPE and prepared SPE/NPC in a 0.1 M KCl solution containing 1.0 mM K_3_[Fe(CN)_6_]/K_4_[Fe(CN)_6_]. (**B**) Stripping curves of the electrode before and after modification in acetic acid buffer solution containing Pb(II) and Cd(II) with a concentration of 30 μg L^−1^ and 0.81 mg L^−1^ of Bi(III).

**Figure 3 foods-12-00730-f003:**
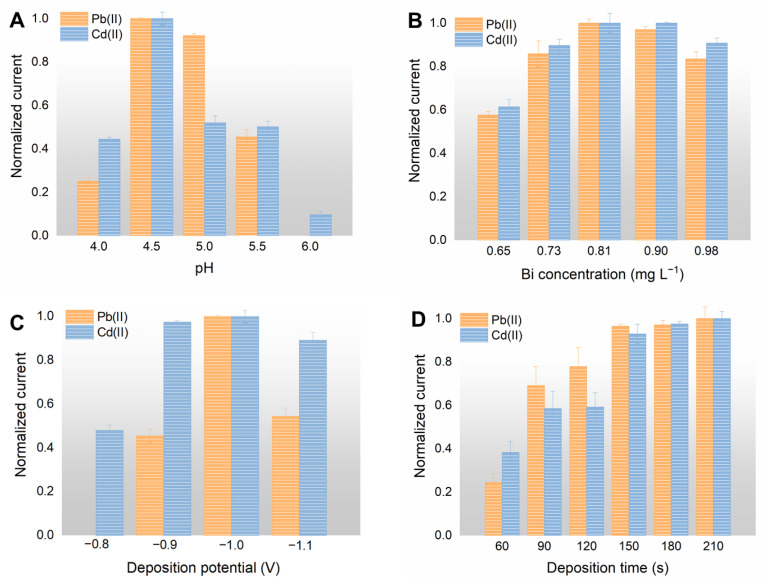
Influences of (**A**) the pH of the acetic acid buffer solution, (**B**) the concentration of Bi(III) ion, (**C**) deposition potential, and (**D**) deposition time on the stripping current of the HMIs.

**Figure 4 foods-12-00730-f004:**
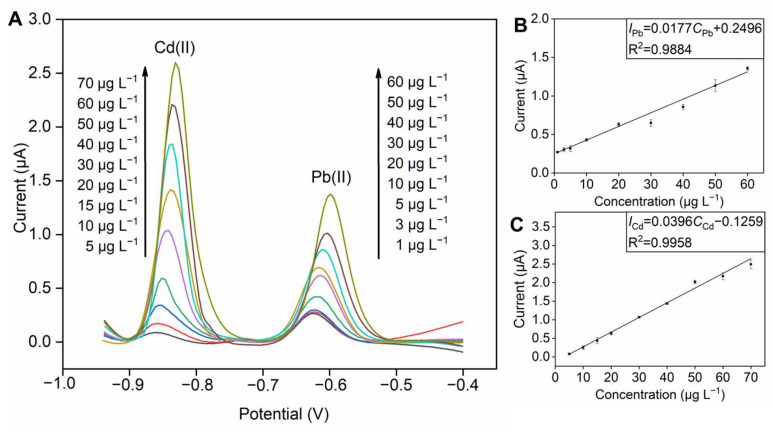
(**A**) Stripping voltammograms of the SPE/NPC for Pb(II) and Cd(II) with various concentrations in acetic acid buffer solution with 0.81 mg L^−1^ Bi(III) and the calibration plots of (**B**) Pb(II) and (**C**) Cd(II).

**Figure 5 foods-12-00730-f005:**
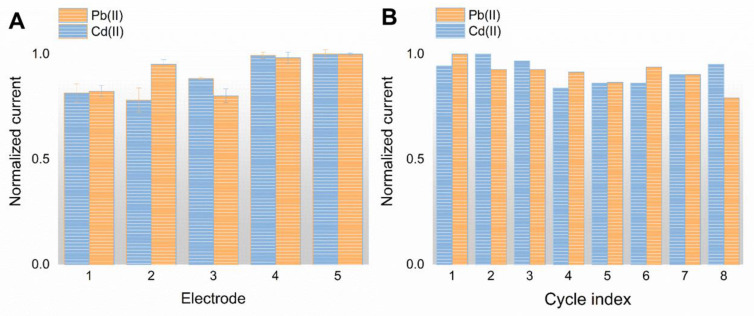
Reproducibility and stability testing results of the SPE/NPC. (**A**) Stripping responses of five SPE/NPC electrodes toward 30 μg L^−1^ of Pb(II) and Cd(II). (**B**) Stripping currents of the SPE/NPC toward 30 μg L^−1^ HMIs after 8 operation cycles.

**Table 1 foods-12-00730-t001:** Recovery measurement for HMI detection in the extracted infant milk powder.

Sample	Cd(II)				Pb(II)			
	Added (μg L^−1^)	ICP-MS (μg L^−1^)	Our Method (μg L^−1^)	Recovery (%)	Added (μg L^−1^)	ICP-MS (μg L^−1^)	Our Method (μg L^−1^)	Recovery (%)
1	0	0	0	/	0	0	0	/
	30	33.23 ± 1.18	21.45 ± 1.38	71.5	30	27.25 ± 3.73	33.12 ± 1.48	110.4
2	0	0	0	/	0	0	0	/
	30	33.19 ± 0.42	25.83 ± 3.54	86.1	30	29.65 ± 0.82	34.06 ± 1.33	113.6
3	0	0	0	/	0	0	0	/
	30	33.24 ± 0.08	27.94 ± 2.88	93.2	30	27.17 ± 0.60	31.05 ± 1.85	103.5
4	0	0	0	/	0	0	0	/
	30	32.19 ± 0.07	23.39 ± 1.38	77.9	30	31.49 ± 1.56	36.13 ± 4.79	120.5
5	0	0	0	/	0	0	0	/
	30	33.57 ± 0.59	31.56 ± 3.90	105.2	30	37.69 ± 0.66	38.58 ± 3.39	128.6

## Data Availability

The data used to support the findings of this study can be made available by the corresponding author upon request.

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
