# Peer review of "Determination of Heavy Metal Ions in Infant Milk Powder Using a Nanoporous Carbon Modified Disposable Sensor"

_foods, 2023, doi:10.3390/foods12040730_

Round 1

Reviewer 1 Report

1.       Pls describe the principle of Nanoporous Carbon Modified Disposable Sensor in detecting Pb(II) and Cd(II), especially the chemical interaction between heavy metal ions and sensor.

2.       Why only Pb(II) and Cd(II) were selected in this study? Other heavy metals were not considered in this work?

3.       Five different brands of infant milk powder were selected in this study. Please list the ingredients/nutrients of these milk powders in table. I believe that it can be observed through the label on packaging of each milk powders.

4.       The reviewer thought Figure 1(C) should refer to Raman spectra, instead of Figure 1(D) (it is more likely XRD spectra).

5.       [line 193] Which is Figure S2?

6.       What mechanism involved that make the surface area of modified electrode higher than bare electrode?

7.       The role played by pH value of acetic acid buffer, deposition volume of NPC sus- 191 pension, concentrations of Bi(III), deposition parameters such as potential and time in improvement of detection performance need to be discussed thoroughly.

8.       Why concentrations of Bi(III) has been considered in optimization? Why Bi(III), instead of Pb(II) and Cd(II), since these are aim of this study?

9.       “Nomalized” appear in label of axis-y in all graphs of Figure 3. It should be “nomalized”or “normalized”. Please clarify.

10.   What are these deposition parameters mean?

11.   What is the stripping performance and stripping current referring to?

12.   [line 246] Figure S4 or Figure 4?

13.   The configuration of electrochemical measurement setup can be presented in a diagram or photo.

14.   Please discuss the factor that lead to the discrepancy of concentration Pb(II) and Cd (II) between ICP-MS and implemented method.

15.   The average recoveries that presented in Section 3.6 and Section 4 are different with the percentage tabulated in Table 1. Please clarify.

16.   Both sets of linear equation (set 1 and set 2) were presented in section 3.4. They have not been described and compared thoroughly. It is confusing. In the meantime, they have not been verified in term of its accuracy in prediction.

IPb = 0.0177CPb + 0.2496, ICd = 0.0396CCd – 0.1259 (set 1)

IPb = 0.0376CPb + 0.3293, ICd = 0.0637CCd – 0.8962.(Set 2)

17.   The significant findings in section 3.5 need to be summarized and highlighted too in Conclusion .

Author Response

We thank the reviewer for the thought-provoking comments, here are our responses.

  1. Pls describe the principle of Nanoporous Carbon Modified Disposable Sensor in detecting Pb(II) and Cd(II), especially the chemical interaction between heavy metal ions and sensor.

Response:The electrochemical detection is normally performed with a three-electrode system containing a working electrode (WE), a reference electrode (RE) and a count electrode (CE) (Figure 1). The WE can be modified with different materials for specific recognition and concentration of metal ions with a specific voltage bias. The presence of heavy metal ions causes the change of current, potential, electrochemical impedance or capacitance, which can then be used for their detection. By optimized functionalization process, nanomaterials can be easily assembled on electrode surface for the sensitive and selective detection of heavy metal ions. The assembly of nanomaterials can further improve the electrochemical performance.

Figure 1. Schematic illustration of general principle of electrochemical sensing of heavy metal ions[1].

The electrochemical detection methods of heavy metal ions involve two basic processes of deposition and stripping (Figure 2). Firstly, a pre-concentration step that involves electro-reduction (deposition) of metal ionic species in the solution to the corresponding zero-valent metal onto WE surface at a constant potential. In the second step, re-oxidation of the zero-valent metals back to the corresponding cations is realized by applying a voltage scan in the anodic direction. The second step causes stripping (dissolution) of the analyte deposited onto the electrode surface at a specific potential. As a result, an extremely fast electrochemical reaction occurs which leads to the production of a strong current that is proportional to the concentration of metal ions present in the medium. Based on the fact that oxidation of each metal requires specific potential [2], the measured values of peak current are proportional to the concentrations of the analyte.

Figure 2. Schematic illustration of the deposition and stripping process in the detection of HMIs[3].

  1. Why only Pb(II) and Cd(II) were selected in this study? Other heavy metals were not considered in this work?

Response:As reported, the main heavy metals of toxic residue in food are Pb, Cd, Hg, As, Ni, Cu, and Zn[4]. Among the family of heavy metals, Pb and Cd are two typical representatives on account of their toxic effects on human living organisms[5]. While Pb is characterized by non-biodegradability, long half-life and producing serious hazards to human health. Cd has a variety of toxic effects on kidney, liver, nerve and cardiovascular system[6]. Therefore, in this study, Pb and Cd were selected as representatives to construct a method for detecting heavy metals in food.

  1. Five different brands of infant milk powder were selected in this study. Please list the ingredients/nutrients of these milk powders in table. I believe that it can be observed through the label on packaging of each milk powders.

Response:The ingredients/nutrients of these milk powders are list in table S2, the table has been added in the manuscript.

Brand

Main ingredients

Nutrients

1

Raw milk, lactose, edible plant blend oil, desalted whey powder, whey protein powder, walnut oil, vitamin A, vitamin D, vitamin E, vitamin K, vitamin B6, nicotinic acid, folic acid, pantothenic acid, vitamin C, biotin,  potassium chloride, copper sulfate, magnesium sulfate, zinc sulfate, sodium selenite, inositol, taurine.

Energy, protein, fat (linoleic acid, α-linolenic acid), carbohydrates, vitamins, minerals.

2

Lactose, whey protein concentrate, skim milk powder, edible plant blend oil, galactose oligosaccharide, minerals (calcium carbonate, magnesium hydrogen phosphate, ferrous sulfate, copper sulfate), vitamins (L-sodium ascorbate, inositol, palmitate vitamin A, nicotinamide, L-ascorbic acid, riboflavin), nucleotides, potassium hydroxide.

Energy, protein, fat (linoleic acid), carbohydrates, vitamins, minerals.

3

Raw milk, skim milk powder, desalted whey powder, lactose, edible plant blend oil, whey protein powder, casein, inositol, taurine, L-carnitine tartrate, nucleotides, phospholipids, vitamin A, vitamin D, vitamin E, vitamin K, vitamin B1, vitamin B2, vitamin B6, nicotinamide, folic acid,  ferrous sulfate, sodium citrate, potassium citrate, potassium chloride.

Energy, protein, fat (linoleic acid, α-linolenic acid), carbohydrates, vitamins, minerals.

4

Raw milk, desalted whey powder, lactose, vegetable oil, whey protein powder, walnut oil, anhydrous cream, phospholipids, vitamin A, vitamin D, vitamin E, vitamin K, vitamin B1, vitamin B6, vitamin B12, nicotinic acid, folic acid, pantothenic acid, vitamin C, copper sulfate, potassium iodate, inositol, taurine, L-carnitine.

Energy, protein, fat (linoleic acid), carbohydrates, vitamins, minerals.

5

Raw milk, desalted whey powder, edible plant blend oil, lactose, galactose oligosaccharides, whey protein concentrate, fructose oligosaccharides, calcium carbonate, magnesium chloride, potassium citrate, dipotassium hydrogen phosphate, sodium selenite, ferric pyrophosphate, zinc sulfate, copper sulfate, manganese sulfate, taurine, vitamin C, vitamin A, vitamin K, nicotinamide, pantothenic acid, inositol, choline.

Energy, protein, fat (linoleic acid, α-linolenic acid), carbohydrates, vitamins, minerals.

  1. The reviewer thought Figure 1(C) should refer to Raman spectra, instead of Figure 1(D) (it is more likely XRD spectra).

Response:This mistake has been corrected in the manuscript. Figure 1(C) refers to Raman spectra, and Figure 1(D) refers to XRD spectra.

  1. [line 193] Which is Figure S2?

Response:Figure S2 (S4 after revised) is in the supporting information. We have added the supporting information to the end of the manuscript.

Figure S4. Effects of deposition volume of NPC solution on the stripping responses of SPE/NPC in the detection of Pb(II) and Cd(II) (30 μg L‒1).

  1. What mechanism involved that make the surface area of modified electrode higher than bare electrode?

Response:Mesoporous materials have been considered as an efficient modifying material owing to its uniform pore sizes, high surface areas, and fast electron transfer rate. Due to the porous structure, the nanoporous carbon owns higher specific surface area, so the surface area of modified electrode is higher than bare electrode. As shown in N2 adsorption-desorption isotherms, the Brunauer-Emmett-Teller (BET) surface area of NPC was 217 m2 g−1, which demonstrates the large specific surface area of the nanoporous carbon.

  1. The role played by pH value of acetic acid buffer, deposition volume of NPC suspension, concentrations of Bi(III), deposition parameters such as potential and time in improvement of detection performance need to be discussed thoroughly.

Response:Here is an optimization of the experimental conditions. In electrochemical detection, buffer is the place where redox reaction occurs, so the pH of buffer is very important. Bi(III) will hydrolyze with high pH, which is detrimental to the depositional process. Therefore, it is necessary to optimize the buffer pH. NPC is the active center of electrode reaction, when the amount of NPC is not enough, the deposition process of heavy metal will be restricted, but too much NPC will form thick film and affect mass transfer. So, it is necessary to optimize it. Moreover, Bi(III) facilitate the deposition process by forming alloys with heavy metal ions, so the concentration of Bi(III) also needs to be optimized. The depositional parameters take important role in deposition process, a systematic optimization work is necessary.

  1. Why concentrations of Bi(III) has been considered in optimization? Why Bi(III), instead of Pb(II) and Cd(II), since these are aim of this study?

Response:Because Bi(III) is the component of the solution under test, the concentration of Bi(III) in the system affects the detection results. The thickness of the Bi-based alloy film is controlled by the concentration of added Bi(III) solution, the electrochemical deposition process of HMIs can be promoted effectively through the formation of Bi-based alloy, however, the mass transfer of metallic deposition process might be hindered by a thick Bi film formed under a high concentration of Bi(III). Therefore, it is necessary to optimize the concentration of Bi(III). Pb(II) and Cd(II) are the detection targets of the experiment, there is no need to optimize.

  1. “Nomalized” appear in label of axis-y in all graphs of Figure 3. It should be “nomalized” or “normalized”. Please clarify.

Response:The spelling mistakes have been revised as “normalized” in Figure 3.

Figure 3. Influences of (A) pH of acetic acid buffer solution, (B) the concentration of Bi(III) ion, (C) deposition potential and (D) deposition time on the stripping current of HMIs.

  1. What are these deposition parameters mean?

Response: In the process of electrochemical deposition, we need to set detection parameters according to the specific testing analytes. Therefore, deposition parameters should be defined according to the actual detection conditions, ensuring an optimized detection method in the HMIs deposition.

  1. What is the stripping performance and stripping current referring to?

Response: In the electrochemical detection process of HMIs, three steps were involved. Firstly, the preconditioning process was operated to clean the electrode surface. Then, the electrochemical deposition of HMIs was performed. Finally, the SWV method was used for the stripping determination of HMIs. At the end of all programs, a stripping curve will be obtained. The stripping performance is the magnitude of the stripping current, which reflects the concentration of the ions to be measured.

  1. [line 246] Figure S4 or Figure 4?

Response: The stripping response diagrams of single variable ions are shown in Figure S4(S5 after revised), and relevant standard curves are attached as well.

Figure S5. Stripping voltammograms of SPE/NPC for single variable of (A) Pb(II) and (B) Cd(II) with various concentrations in acetic acid buffer solution with 0.81 mg L‒1 Bi(III), and the calibration plots of (C) Pb(II) and (D) Cd(II).

  1. The configuration of electrochemical measurement setup can be presented in a diagram or photo.

Response: The configuration of electrochemical measurement setup has been presented in the diagram below.

  1. Please discuss the factor that lead to the discrepancy of concentration Pb(II) and Cd(II) between ICP-MS and implemented method.

Response: There are three factors that lead to the discrepancy of concentration Pb(II) and Cd(II) between ICP-MS and implemented method. First, the form of the ion affects the test results. In the samples, heavy metal ions mainly exist in free state, but a few exist in bound state. The electrochemical method can only detect ions in the free state, while the ICP-MS method can detect both free and bonded ions due to the addition of strong acids. The second point is the matrix effect of different milk powder. Since the standard curve is established in the buffer, there will be some deviation when it is applied to different milk powder due to the different sensitivity. The last is the measurement error inherent in the ICP-MS method, which may be caused by instrument, volumetric bottle and human operation.

  1. The average recoveries that presented in Section 3.6 and Section 4 are different with the percentage tabulated in Table 1. Please clarify.

Response: In Section 3.6, the data here (115.32% and 86.78%) is the average of the data in the table 1, which includes the two detected ions. In Section 4, the data here is repeated. The behavior of single variable heavy metal ions in real samples are test as well in Section 3.6. The data here (83.08% and 102.76%) is the average of the data in the table S4. And this situation is for one ion detected alone and opposed to the situation above where we measure two ions together.

  1. Both sets of linear equation (set 1 and set 2) were presented in section 3.4. They have not been described and compared thoroughly. It is confusing. In the meantime, they have not been verified in term of its accuracy in prediction.

IPb = 0.0177CPb + 0.2496, ICd = 0.0396CCd – 0.1259 (set 1)

IPb = 0.0376CPb + 0.3293, ICd = 0.0637CCd – 0.8962.(Set 2)

Response: Set 1 is the linear equation obtained when two heavy metal ions are detected simultaneously, and Set 2 is the linear equation obtained when heavy metal ions are detected separately. Their prediction accuracy has been verified by standard addition method, and corresponding results are shown in Table 1 and Table S4. In the table, the predicted results were compared with those obtained by standard detection methods, and high recovery rates was obtained.

  1. The significant findings in section 3.5 need to be summarized and highlighted too in Conclusion.

Response: It has been summarized and highlighted in Conclusion.

After a long-term operation, no obvious change in the stripping responses of HMIs was observed. For the selectivity evaluation, acceptable changes in the peak current compared with the responses of Pb(II) and Cd(II) were observed after the addition of interferences. Though excellent stability and anti-interference performance were achieved, the reproducibility still needs to be further improved.

References

  1. Cui, L.; Wu, J.; Ju, H. Electrochemical sensing of heavy metal ions with inorganic, organic and bio-materials. Biosens Bioelectron. 2015, 63, 276-286.
  2. Waheed, A.; Mansha, M.; Ullah, N. Nanomaterials-based electrochemical detection of heavy metals in water: Current status, challenges and future direction. TrAC Trends in Analytical Chemistry. 2018, 105, 37-51.
  3. Liu, X.; Yao, Y.; Ying, Y.; Ping, J. Recent advances in nanomaterial-enabled screen-printed electrochemical sensors for heavy metal detection. TrAC Trends in Analytical Chemistry. 2019, 115, 187-202.
  4. Wang, L.; Peng, X.; Fu, H.; Huang, C.; Li, Y.; Liu, Z. Recent advances in the development of electrochemical aptasensors for detection of heavy metals in food. Biosens Bioelectron. 2020, 147, 111777.
  5. Yao, Y.; Wu, H.; Ping, J. Simultaneous determination of Cd(II) and Pb(II) ions in honey and milk samples using a single-walled carbon nanohorns modified screen-printed electrochemical sensor. Food Chem. 2019, 274, 8-15.
  6. Li, G.; Belwal, T.; Luo, Z.; Li, Y.; Li, L.; Xu, Y.; Lin, X. Direct detection of Pb(2+) and Cd(2+) in juice and beverage samples using PDMS modified nanochannels electrochemical sensors. Food Chem. 2021, 356, 129632.

Reviewer 2 Report

[1] Why not the authors have used Adsorptive Stripping Square Wave Voltammetry rather than Square Wave Voltammetry? It is desired and must for stripping.

[2] Provide FRA and BET analysis of synthesized material.

[3] Author must provide in the detail literature comparison Table regarding of detection of lead (II) and cadmium (II) by electrochemical and other widely used techniques.

[4] In the figure 4 A, the potential shift observed for same metal is in very wide range, origin of all the peaks is very different. They must start from the same point.

[5] I do not find any novelty from this article. 

Author Response

We thank the reviewer for the thought-provoking comments, here are our responses.

Line 103., a small amount of NPC bulk powder was dissolved in the mixture of ethanol and water…” Please define the preparation of the solution more precisely.

Response:It has been defined more precisely in the manuscript. To obtain homogeneous solution of NPC, 0.02g of NPC bulk powder was dissolved in 10 mL of ethanol and 10 mL of water with sonication for 30 min.

Line 116. „Specifically, the milk powder was dissolved completely into DI water, then, 50 μL of H2O2 (30 wt%) was added into… ” Please define the sample pretreatment more precisely.

Response: It has been defined more precisely in the manuscript.

Specifically, to ensure the complete dissolution of milk powder, 2.87 g milk powder was added to 20 mL DI water. Then 50 µL of H2O2 (30 wt%) was added into 20 mL of milk to remove the reductive substance like vitamins.

Line 128. „Bi(III) was added before the HMIs detection and sample extractions for promoting the HMIs reduction by forming alloy.” Please describe more detailed the measuring cell or the arrangement of the measurement.

Response: It has been added in the manuscript.

The testing well was filled with acetic acid buffer first. Bi(NO3)3 was then added to form alloy with HMIs and facilitate the deposition process. After adding Pb(II) and Cd(II) into the mixing solution, the electrochemical program of CHI work station was operated.

Line 151. Fig 1C and D are changed compared to the description.

Response:It has been corrected in the manuscript.

Figure 1. (A, B) Scanning electron microscope images, (C) Raman spectra of NPC, and (D) X-ray diffraction pattern.

Line 204. Fig3 C. The plot should be presented between -0.4 and -1.1 V to show both ions measured.

Response:When deposition potential is lower than or equal to -0.8 V, the stripping current of Pb(II) is nearly zero. In order to realize simultaneous detection of two kinds of ions, the optimization range from -0.8V to -1.1V is selected during potential optimization.

Line 230. Fig3 D. „As for the deposition potential (Figure 3D)” Please correct it.

Response:It has been corrected in the manuscript.

As for the deposition potential (Figure 3C), Pb(II) and Cd(II) were barely detected with the potential of −0.8 V. From −0.8 V to −1.0 V, the signal begins to strengthen. With the deposition potential exceeding −1.0 V, the current responses begin to decrease due to the reduction of interfering substances in the solution under high potential. From the results exhibited in Figure 3D, the current response has an obvious increasing trend as the deposition time prolongs, and then it gradually approaches a flat after 150 s of deposition, which can be explained as the HMIs deposited on the electrode have reached saturation. Accordingly, a depositing time of 150 s at a deposition voltage of −1.0 V was applied in the end.

Line 263. „Besides, the stability of SPE/NPC was also evaluated by a long-term operation. The peak currents of SPE/NPC are recorded in eight consecutive detection times” Please define it more precisely.

Response:It has been defined more precisely in the manuscript.

Besides, the stability of SPE/NPC was also evaluated by a long-term operation, which means a same electrode was used eight times to measure Pb(II) and Cd(II). The peak currents of SPE/NPC were recorded in eight consecutive detection times.

Line 286. “A recovery test was used to assess the performance of our method for real sample analysis.” Why were the standards added to the pretreated samples and not to the original solution? There may be a risk that the concentration decreases e.g. during the deproteinization.

Response:The pretreatment process is to remove the interference substances that may exist in milk powder and affect the detection of heavy metal ions, mainly reducing substances and bio-macromolecules. Here, oxidants and acids are mainly added and heavy metal ions don’t react with them. In the process of deproteinization, due to the addition of hydrochloric acid and acetic acid, the pH of the system is low enough, heavy metals exist in the form of ions, and the loss in the process of protein removal by filtration is very small. The treatment adopted in this paper mainly refers to the method of Ping et al[1]. In the detection of heavy metal ions in some other foods, different pretreatment methods are also used[2, 3].

References

  1. Ping, J.; Wu, J.; Ying, Y. Determination of trace heavy metals in milk using an ionic liquid and bismuth oxide nanoparticles modified carbon paste electrode. Chinese Science Bulletin. 2012, 57, 1781-1787.
  2. Silva, L.P.; Campos, N.D.S.; Lisboa, T.P.; de Faria, L.V.; Matos, M.A.C.; Matos, R.C.; de Sousa, R.A. Simultaneous determination of cadmium, lead and copper in chocolate samples by square wave anodic stripping voltammetry. Food Addit Contam Part A Chem Anal Control Expo Risk Assess. 2021, 38, 418-426.
  3. Li, G.; Belwal, T.; Luo, Z.; Li, Y.; Li, L.; Xu, Y.; Lin, X. Direct detection of Pb(2+) and Cd(2+) in juice and beverage samples using PDMS modified nanochannels electrochemical sensors. Food Chem. 2021, 356, 129632.

Reviewer 3 Report

I was happy to read about the electrochemical method developed for measuring heavy metal pollution in formulas. The solution's simplicity and detection parameters make it suitable for practical application.

However, I have a few comments about the manuscript:

Line 103. „a small amount of NPC bulk powder was dis solved in the mixture of ethanol and water…” Please define the preparation of the solution more precisely.

Line 116. „Specifically, the milk powder was dissolved completely into DI water, then, 50 μL of H2O2 (30 wt%) was added into… ” Please define the sample pretreatment more precisely.

Line 128. „Bi(III) was added before the HMIs detection and sample extrac- 129 tions for promoting the HMIs reduction by forming alloy.” Please describe more detailed the measuring cell or the arrangement of the measurement.

Line 151. Fig 1C and D are changed compared to the description.

Line 204. Fig3 C. The plot should be presented between -0.4 and -1.1 V to show both ions measured.

Line 230. Fig3 D. „As for the deposition potential (Figure 3D)” Please correct it.

Line 263. „Besides, the stability of SPE/NPC was also evaluated by a long-term operation. The peak currents of SPE/NPC are recorded in eight consecutive 264 detection times” Please define it more precisely.

Line 286. “A recovery test was used to assess the performance of our method for real sample analysis.” Why were the standards added to the pretreated samples and not to the original solution? There may be a risk that the concentration decreases e.g. during the deproteinization.

Author Response

We thank the reviewer for the thought-provoking comments, here are our responses.

[1] Why not the authors have used Adsorptive Stripping Square Wave Voltammetry rather than Square Wave Voltammetry? It is desired and must for stripping.

Response: We have replaced Square Wave Voltammetry with Adsorptive Stripping Square Wave Voltammetry in the full text.

[2] Provide FRA and BET analysis of synthesized material.

Response: BET and FRA analysis of synthesized material was added in Figure S1 and Figure S2.

N2 adsorption−desorption isotherms were measured to investigate the pore structure of the NPC material (Figure S1). The Brunauer−Emmett−Teller (BET) surface area of NPC was 217 m2 g−1. As observed in the pore-size distribution, the main pore size of NPC was 26.42 nm.

We also performed EIS analysis to measure the impedance of the NPC. Electrochemical impedance spectroscopy (EIS) was used to characterize the nanomaterials modification of the sensor (Figure S2). The diameter of the high-frequency semicircle in Nyquist spectra represents the electron transfer resistance (RET) at the electrode surface. When the surface of the SPE was coated with NPC, the RET value decreased from 2106 Ω to 1806 Ω, indicating that the presence of NPC could promote the electron transfer at the electrode interface.

[3] Author must provide in the detail literature comparison Table regarding of detection of lead (II) and cadmium (II) by electrochemical and other widely used techniques.

Response:It has been provided here and added in the manuscript as Table S1.

Principle

Characteristics

References

Spectroscopy

Atomic Absorption Spectroscopy

The absorption degree of gaseous ground state atoms to characteristic spectral lines is measured.

High sensitivity, low detection limit, little background interference. However, the pretreatment is complicated, the equipment is large and the operation is tedious.

 [1-3]

Atomic Emission Spectroscopy

Qualitative and quantitative analysis is carried out by measuring the characteristic spectrum of the electron transition in the outer layer of the atom under the state of electric or thermal excitation.

High sensitivity, good selectivity, fast detection speed. But the equipment is complicated and the error is large.

Atomic Fluorescence Spectrometry

The element under test is determined by measuring the emission intensity of fluorescence produced by atomic vapor of the element under radiation excitation.

The spectrum line is simple and less interference, and the linear range is wide. But the application elements are limited.

Mass Spectrometry

Inductively Coupled Plasma Mass Spectrometry

The inductively coupled plasma is combined with mass spectrum, the sample is vaporized by inductively coupled plasma, the metal to be measured is separated, and then the mass spectrum is used for determination.

It can be used for qualitative analysis, semi-quantitative analysis and quantitative analysis, and can be used for the determination of various elements and isotopes at the same time. The detection limit is low. But they are expensive and vulnerable to contamination.

 [4, 5]

Chromatography

High Performance Liquid Chromatography

Trace metal ions form stable colored complexes with organic reagents, which are separated by liquid chromatography and detected by UV-vis detector.

Simultaneous determination of multiple elements can be realized. However, the selection of complex reagents is limited, which brings limitations to the wide application of liquid chromatography.

   [6]

Biochemical technology

Enzyme Inhibition

The toxicity of heavy metal ions can be used to reduce the enzyme activity after the combination with enzymes,  the color, electrical conductivity, absorbance and other signals change, thus reflecting the type or content of heavy metals.

The detection speed is fast and the operation is simple. But the enzyme type is limited and the specificity is poor.

[7-11] 

Immunoassay

The specific reaction of heavy metal antigen and monoclonal antibody was used for detection.

Strong specificity. But the cost is high and it is impossible to detect multiple elements at the same time.

Electrochemical method

(DPASV, SWASV, LSASV)

The electrochemical activity of the test substance is used and the electrical signal is transformed to realize qualitative and quantitative detection.

High sensitivity, fast detection speed, the ability to measure a variety of substances at the same time, easy to operate, portable instrument.

 [12-15]

DPASV: differential pulse anodic stripping voltammetry; SWASV: square wave anodic stripping voltammetry; LSASV linear sweep anoding stripping voltammetry.

[4] In the figure 4 A, the potential shift observed for same metal is in very wide range, origin of all the peaks is very different. They must start from the same point.

Response:Test environment and stability of reference electrode affect potential drift. Here, corresponding potential of the peak current of Cd(II) is from -0.86 V to -0.83 V, and that of Pb(II) is from -0.62 V to -0.59 V.

[5] I do not find any novelty from this article. 

Response:As a new type of carbon nanomaterial, notable merits of low density, excellent thermal and chemical stability, hierarchical porosity, efficient mass transport, and large adsorption capacity make NPC exhibits satisfactory properties in the adsorption, catalyst, and energy storage application[16, 17]. Most importantly, no reports explored the performance of NPC in the electrochemical detection of HMIs until now. In this work, SPE and NPC were combined and the prepared disposable SPE/NPC with excellent stability and anti-interference performance in the electrochemical detection of HMIs process, it’s an innovative combination and application.

References

  1. Pan, F.; Yu, Y.; Yu, L.; Lin, H.; Wang, Y.; Zhang, L.; Pan, D.; Zhu, R. Quantitative assessment on soil concentration of heavy metal-contaminated soil with various sample pretreatment techniques and detection methods. Environ Monit Assess. 2020, 192, 800.
  2. Li, K.; Yang, H.; Yuan, X.; Zhang, M. Recent developments of heavy metals detection in traditional Chinese medicine by atomic spectrometry. Microchemical Journal. 2021, 160, 105726.
  3. Arduini, F.; Palleschi, G. Screening and confirmatory methods for the detection of heavy metals in foods. Persistent Organic Pollutants and Toxic Metals in Foods. 2013, 81-109.
  4. Zhang, J.; Chen, B.; Wang, H.; He, M.; Hu, B. Facile Chip-Based Array Monolithic Microextraction System Online Coupled with ICPMS for Fast Analysis of Trace Heavy Metals in Biological Samples. Anal Chem. 2017, 89, 6878-6885.
  5. Yu, X.; Chen, B.; He, M.; Wang, H.; Tian, S.; Hu, B. Facile Design of Phase Separation for Microfluidic Droplet-Based Liquid Phase Microextraction as a Front End to Electrothermal Vaporization-ICPMS for the Analysis of Trace Metals in Cells. Anal Chem. 2018, 90, 10078-10086.
  6. Burakham, R.; Srijaranai, S.; Grudpan, K. High-performance liquid chromatography with sequential injection for online precolumn derivatization of some heavy metals. J Sep Sci. 2007, 30, 2614-9.
  7. Wang, Y.; Zhang, C.; Liu, F. Antibody developments for metal ions and their applications. Food and Agricultural Immunology. 2020, 31, 1079-1103.
  8. Ouyang, H.; Shu, Q.; Wang, W.; Wang, Z.; Yang, S.; Wang, L.; Fu, Z. An ultra-facile and label-free immunoassay strategy for detection of copper (II) utilizing chemiluminescence self-enhancement of Cu (II)-ethylenediaminetetraacetate chelate. Biosens Bioelectron. 2016, 85, 157-163.
  9. Date, Y.; Terakado, S.; Sasaki, K.; Aota, A.; Matsumoto, N.; Shiku, H.; Ino, K.; Watanabe, Y.; Matsue, T.; Ohmura, N. Microfluidic heavy metal immunoassay based on absorbance measurement. Biosens Bioelectron. 2012, 33, 106-12.
  10. Compagnone, D.; Lupu, A.S.; Ciucu, A.; Magearu, V.; Cremisini, C.; Palleschi, G. Fast Amperometric Fia Procedure for Heavy Metal Detection Using Enzyme Inhibition. Analytical Letters. 2001, 34, 17-27.
  11. Bachan Upadhyay, L.S.; Verma, N. Enzyme Inhibition Based Biosensors: A Review. Analytical Letters. 2013, 46, 225-241.
  12. Wang, X.; Qi, Y.; Shen, Y.; Yuan, Y.; Zhang, L.; Zhang, C.; Sun, Y. A ratiometric electrochemical sensor for simultaneous detection of multiple heavy metal ions based on ferrocene-functionalized metal-organic framework. Sensors and Actuators B: Chemical. 2020, 310, 127756.
  13. Wang, L.; Peng, X.; Fu, H.; Huang, C.; Li, Y.; Liu, Z. Recent advances in the development of electrochemical aptasensors for detection of heavy metals in food. Biosens Bioelectron. 2020, 147, 111777.
  14. Nantao, L.; Diming, Z.; Qian, Z.; Yanli, L.; Jing, J.; Gang Logan, L.; Qingjun, L. Combining localized surface plasmon resonance with anodic stripping voltammetry for heavy metal ion detection. Sensors and Actuators B: Chemical. 2016, 231, 349-356.
  15. Bansod, B.; Kumar, T.; Thakur, R.; Rana, S.; Singh, I. A review on various electrochemical techniques for heavy metal ions detection with different sensing platforms. Biosens Bioelectron. 2017, 94, 443-455.
  16. Chung, D.Y.; Son, Y.J.; Yoo, J.M.; Kang, J.S.; Ahn, C.Y.; Park, S.; Sung, Y.E. Coffee Waste-Derived Hierarchical Porous Carbon as a Highly Active and Durable Electrocatalyst for Electrochemical Energy Applications. ACS Appl Mater Interfaces. 2017, 9, 41303-41313.
  17. Vilian, A.T.E.; Song, J.Y.; Lee, Y.S.; Hwang, S.K.; Kim, H.J.; Jun, Y.S.; Huh, Y.S.; Han, Y.K. Salt-templated three-dimensional porous carbon for electrochemical determination of gallic acid. Biosens Bioelectron. 2018, 117, 597-604.

Round 2

Reviewer 2 Report

I have through the revised manuscript thoroughly and the authors have written and included proper references. So my decision is Accept the article.

Reviewer 3 Report

The answers and additions to my questions are adequate and satisfactory.

I recommend accepting the manuscript in its current form.